# Promoting sparsity in continuous-time models to learn delayed dependence structures

## Abstract

Continuous-time dynamics models, such as neural ordinary differential equations, enable accurate modeling of underlying dynamics in time-series data. However, the use of neural networks in parameterizing dynamics makes it challenging for humans to identify dependence structures, especially in the presence of *delayed* effects. In consequence, these models are not an attractive option when capturing dependence carries more importance than accurate predictions, *e.g.*, tsunami forecasting. In this paper, we present a novel method for learning dependence structures in continuous-time dynamics models. Inspired by neural graphical modeling, we promote weight sparsity in the network's first layer during training. Once trained, we prune the sparse weights to identify dependence structures. In evaluation, we first test our method in scenarios where the exact dependence-structures of time-series are known. Our method captures the underlying dependence structure precisely even when there is a delayed effects. We further evaluate our method to a real-world tsunami forecasting, where the exact dependence structures are unknown. Even in this challenging case, our method effective learns physically-consistent dependence structures with a high forecasting accuracy.

## 1 Introduction

An emerging paradigm for modeling underlying dynamics in time-series data is to use continuous-time dynamics neural networks, such as neural ordinary differential equations (NODEs) (Chen et al., 2018; Rubanova et al., 2019; Dupont et al., 2019). Widely known as a continuous-depth extension of residual networks (He et al., 2016), NODEs have a great fit for data-driven dynamics modeling as they construct models in the form of ODEs. This new paradigm has enabled breakthroughs in many applications, e.g., healthcare (Rubanova et al., 2019), computational physics (Lee & Parish, 2021; Lee et al., 2021a;b), or climate modeling (Hwang et al., 2021; Park et al., 2020).

A promising approach to enhance modeling performance is to increase the *expressivity* of the neural network used to parameterize a system, *e.g.*, by leveraging convolutional neural networks or by augmenting extra dimension in the state space (Dupont et al., 2019). While shown effective, as the models are getting more complex, interpreting the learned dynamics becomes more challenging for humans. This poses a greater challenge for tasks that need an interpretable *dependence* structure, e.g., predicting high-consequence events like tsunamis.

In this work, we propose a novel method for identifying human-interpretable dependence structure in time-series using continuous-time neural networks. We focus on a score-based structure learning algorithm for dynamical systems—a framework recently proposed by Bellot et al. (2022) that allows us to quantify the impact of a past event on the future evolution of data in a continuous-time setting. In particular, we extend the score-based structure learning algorithm to the dynamical systems governed by neural delay differential equations (NDDEs) (Zhu et al., 2020). This adaptation enables the identification of a dependence structure of not only the contemporaneous variables, but also the delayed (or lagged) variables.

**Contributions.** *First*, we propose a novel method for extracting the dependence structures from the parameters of a continuous-time neural network with delayed variables. Specifically, we focus on models used for forecasting. To this end, we adapt the score-based structure learning algorithm presented by Bellot et al. (2022) to NDDEs. On the graphical representation of contemporaneous and delayed variables, we enforce sparsity to edges so that edges of less dependence can be eliminated.

This is achieved by using a training algorithm, which minimizes a data-matching loss along with a sparsity-promoting penalty and prunes the columns of input layer weights (i.e., edges) that has smaller norms than a threshold. At the end of the training, the norms of columns corresponding to the important input elements will have high magnitudes, which will make interpretation on structure easier.

*Second*, we test our method on simulated data of chaotic ODEs to see (1) if the model is capable of learning complex chaotic dynamics and (2) if the learned dependence structures match with the known ground-truth structures. We use two continuous-time dynamics models, NODEs and NDDEs, trained on the data sampled from the Lorenz-96 and Mackey–Grass systems, respectively. The results demonstrate that our mechanism precisely extracts the structures from the data.

*Third*, we further evaluate our method in tsunami forecasting at the Strait of Juan de Fuca (Melgar et al., 2016), where we do not know the exact dependence structures in the data. We train ND-DEs on the tsunami dataset (Melgar, 2016) by using the proposed training algorithm. We show that our method outperforms the recent deep-feedfoward networks-based baselines proposed in the prior work (Liu et al., 2021) and effective in predicting the highest peaks of sea-surface elevations at locations near highly-populated areas. We further show that one can approximately capture the physically-consistent dependence structures between a tsunami event and tide observed before, which aligns with the domain expert's interpretation of the same phenomenon.

## 2 PRELIMINARIES

**Neural ordinary differential equations (NODEs)** parameterize the time-continuous dynamics of hidden states in the data as a system of ODEs using a neural network (Chen et al., 2018):

$$\frac{d\boldsymbol{z}}{dt} = \boldsymbol{f}(\boldsymbol{z}, t; \boldsymbol{\Theta}) \tag{1}$$

where $\boldsymbol{z}(t) \in \mathbb{R}^{n_z}$ denotes a time-continuous hidden state, $\boldsymbol{f}$ is a velocity function, parameterized by a neural network whose parameters are denoted as $\boldsymbol{\Theta}$. A typical parameterization of $f$ is a multi-layer perceptron (MLP) $\boldsymbol{\Theta} = \{(W^\ell, \boldsymbol{b}^\ell)\}_{\ell=1}$. $\boldsymbol{\Theta}^l$ and $\boldsymbol{b}^l$ are weights and biases of the $\ell$th layer, respectively. The forward pass of NODEs is equivalent to solving an initial value problem via a black-box ODE solver. Given an initial condition $\boldsymbol{z}_0$ and $\boldsymbol{f}$, it computes:

$$\boldsymbol{z}(t_0), \boldsymbol{z}(t_1) = \text{ODESolve}(\boldsymbol{z}_0, \boldsymbol{f}, \{t_0, t_1\}; \boldsymbol{\Theta}). \tag{2}$$

**Neural delay differential equations (NDDEs).** NODEs serve as great means for data-driven dynamics modeling; however, they have several limitations. One obvious limitation in the context of dependence structure modeling is that NODEs take only the current state of the input variables $\boldsymbol{z}(t)$ (shown in Eqn. 1), which is not suitable for capturing delayed causal effects.

To address this challenge, we use the computational formalism provided by NDDEs (Zhu et al., 2020). NDDEs are an extension of NODEs which takes extra input variables $\boldsymbol{z}_{\leq\tau}(t)$ as follows:

$$\frac{d\boldsymbol{z}}{dt} = \boldsymbol{f}(\boldsymbol{z}(t), \boldsymbol{z}_{\leq\tau}(t), t; \boldsymbol{\Theta}), \tag{3}$$

where $\boldsymbol{z}_{\leq\tau}(t) = \{\boldsymbol{z}(t-\gamma) : \gamma \in [0, \tau]\}$ denotes the trajectory of the solution in the past up to time $t-\tau$. To avoid numerical challenges in handling a continuous form of delay, we use a discrete form:

$$\frac{d\boldsymbol{z}}{dt} = \boldsymbol{f}(\boldsymbol{z}(t), \boldsymbol{z}(t-\tau_1), \dots, \boldsymbol{z}(t-\tau_m), t; \boldsymbol{\Theta}), \tag{4}$$

where $m$ indicates the number of discrete delayed variables. While NDDEs are universal approximators (Zhu et al., 2020), most studies use this continuous-depth neural networks for performing standard classification tasks, e.g., visual recognition (Zhu et al., 2020; 2022). A few studies use ND-DEs for time-series modeling, but they are limited to simulated ODE data (Holt et al., 2022; Monsel et al., 2023). It is also unknown whether we can identify dependence structures from these models. To our best knowledge, we are the first who leverage NDDEs for modeling real-world time-series data and discovering *delayed* dependence structures.

**Neural graphical modeling for dependence structure discovery.** In recent years, there have been advances in structure discovery in continuous-time, such as neural graphical modeling (NGM) (Bellot et al., 2022), the score-based structure learning for dynamical systems.

We start from rewriting the system of ODEs in Eqn. 1 as:

$$\frac{dz_j}{dt} = f_j(\boldsymbol{z}, t; \theta_j), \ j = 1, \ldots, n_z \tag{5}$$

where $z_j$ denotes the $j$th element of the hidden state such that $\boldsymbol{z} = [z_1, \ldots, z_{n_z}]^\mathsf{T}$. The velocity function $\boldsymbol{f}$ consists of $n_z$ individual neural networks such that $\boldsymbol{f}(\boldsymbol{z}) = [f_1(\boldsymbol{z}), \ldots, f_{n_z}(\boldsymbol{z})]^\mathsf{T}$ with $f_j \in \mathbb{R}$. Each networks are parameterized by so called model parameters $\theta_j$ (i.e., $\boldsymbol{\Theta} = \cup_{j=1}^{n_z} \theta_j$).

Assuming the typical parameterization (i.e., affine transformation followed by nonlinearity), each individual function $f_j$ can be written as follows (we omit the bias terms for clarity):

$$f_j(\boldsymbol{z}, t; \theta_j) = W_j^L \sigma(\cdots \sigma(W_j^2 \sigma(W_j^1 \boldsymbol{z})) \cdots), \ j = 1, \ldots, n_z, \tag{6}$$

where $\sigma$ is the nonlinear activation, and $W_j^l$ is the weight of the $l$th layer of the $j$th neural network.

Given the dynamics model in Eqn. 5, the (in-)dependence structure within the processes $\boldsymbol{z}$ can be identified via the definition of *local independence* and graphical representations of the processes. Here, we briefly recap the definition of *local independence* (we refer readers to the reference (Bellot et al., 2022) for the formal definition); a process $z_i$ is locally independent of $z_j$ given $z_k$, if the past of $z_k$ up until time $t$ gives the same information for predicting $z_i(t)$ ($z_i$ at time $t$) as the past of $z_i$ and $z_j$ until time $t$. This independence structure may be represented as a directed graph $\mathcal{G} = (V, E)$, where $V$ denotes a set of vertices representing distinct processes $\{z_j\}_{j=1}^{n_z}$ and $E$ denotes a set of edges, where a directed edge $z_i \to z_j \in E$ if and only if $z_i$ is not locally conditionally independent $z_j$.

With local independence and the graph $\mathcal{G}$, here we briefly restate Lemma 1 in Bellot et al. (2022) (Proposition 3.6 in MOGENSEN & HANSEN (2020)); given dynamics models in the form of Eqn. 5, two processes $z_i$ and $z_j$ are locally dependent if and only if $z_i$ appears in the differential equation of $z_j$ (i.e., $||\frac{\partial f_j}{\partial z_i}||_{L_2} \neq 0$, where $|| \cdot ||_{L_2}$ is the functional $L_2$ norm). Moreover, for any $\boldsymbol{f}'$ such that $||f_j' z_i||_{L_2} = 0$, there exists an equivalent vector field $\boldsymbol{f}$ such that the $i$th column of its input layer has the Euclidean norm zero, i.e., $||[W_j^1]_i||_2 = 0$. For the graph model $\mathcal{G}$, we can define the adjacency matrix $\boldsymbol{A} \in \{0, 1\}^{n_z \times n_z}$, where $\boldsymbol{A}_{ij} \neq 0$ if and only if $||f_j z_i||_{L_2} \neq 0$, suggesting the dependence structure can be identified through the input layer parameters $W_j^1$, $j = 1, \ldots, n_z$. Thus, sparsity-promoting regularizes such as the group lasso (Hastie, 2009) or the adaptive group lasso (Holt et al., 2022) can be applied to the input layer parameters $W_j^1$, leading to the identification of the dependence structure (e.g., Fig. 9(a) in Appendix for graphical illustration).

## 3 NEURAL GRAPHICAL MODELING FOR NDDES

Here we present our method for (delayed) dependence structure discovery. NGM demonstrated its effectiveness in finding dependence structures of the contemporaneous variables (those in the same time index), but it has not been shown how we adapt NGM when the system has delayed effects. Our method overcomes this limit and offers a way to characterize delayed structure(s).

### 3.1 (DELAYED) DEPENDENCE STRUCTURE IN CONTINUOUS-TIME MODELS

We find that NDDEs' velocity function (Eqn. 4) has the same structure as those in NODEs (Eqn. 5):

$$\boldsymbol{f}(\bar{\boldsymbol{z}}(t); \boldsymbol{\Theta}) = \begin{bmatrix} f_1(\boldsymbol{z}(t), \boldsymbol{z}(t - \tau_1), \ldots, \boldsymbol{z}(t - \tau_m); \theta_1) \\ \vdots \\ f_{n_z}(\boldsymbol{z}(t), \boldsymbol{z}(t - \tau_1), \ldots, \boldsymbol{z}(t - \tau_m); \theta_{n_z}) \end{bmatrix}, \tag{7}$$

where $f_j \in \mathbb{R}$ is the $j$th element of the velocity function, *i.e.*, $\frac{d\boldsymbol{z}}{dt} = f_j$. The input layer of the $j$th velocity element, $f_j(\boldsymbol{z}) = W_j^1 \bar{\boldsymbol{z}} + \boldsymbol{b}_j^1$. $\bar{\boldsymbol{z}}(t)$ is a vertical concatenation of all delayed variables:

$$\begin{aligned} \bar{\boldsymbol{z}}(t) &= [\boldsymbol{z}(t)^\mathsf{T}, \boldsymbol{z}(t - \tau_1)^\mathsf{T}, \ldots, \boldsymbol{z}(t - \tau_m)^\mathsf{T}] \\ &= [z_1(t), \ldots, z_{n_z}(t), z_1(t - \tau_1), \ldots, z_1(t - \tau_m), \ldots]. \end{aligned} \tag{8}$$

Inspired by the prior work's approach (Bellot et al., 2022), we propose to identify the dependence structure through the input layer parameters $W_j^1$; if the $i$th column of the first layer weight, $W_j^1$,

---

**Algorithm 1** Neural Graphical Modeling in NDDEs

---
Initialize $\Theta$
**for** $(i = 0; \; i < n_{\max}; \; i = i + 1)$ **do**
    Sample $n_{\text{batch}}$ trajectories randomly from $\mathcal{D}_{\text{train}}$
    Sample initial points randomly from the sampled trajectories: $\bar{z}^r(s(r))$, $s(r) \in [0, \ldots, m - \ell_{\text{batch}} - 1]$ for $r = 1, \ldots, n_{\text{batch}}$
    $\tilde{z}(t_1), \ldots, \tilde{z}(t_m) = \text{ODESolve}(\bar{z}^r_{s(r)}, \boldsymbol{f}, \{t_1, \ldots, t_m\}; \boldsymbol{\Theta})$, for $r = 1, \ldots, n_{\text{batch}}$
    Compute the loss (Eqn. 9)
    Update $\boldsymbol{\Theta}$ via Adam
    Prune $\boldsymbol{\Theta}$ based on the magnitude (Eqn. 10)
**end for**

---

contains only zero elements (i.e., $\|[W_j^1]_i\|_2 = 0$), the $j$th element in $\bar{z}(t)$ is independent from the $i$th element of $\bar{z}$. Through this process, we are able to reveal the structure in terms of the adjacency matrix $A \in \mathbb{R}^{n_z(m+1) \times n_z}$ (a graphical illustration of this process is shown in Fig. 9(b) in Appendix).

### 3.2 SPARSITY-PROMOTING LOSS AND PRUNING

We now propose to minimize the following sparsity promoting loss (lasso), and by doing so, we aim to identify the structures from the multivariate time-series over the course of the training:

$$\sum_{i=1}^{n_{\text{train}}} L(z(t_i), \tilde{z}(t_i)) + \alpha \sum_{i=1}^{n_z} \sum_{j=1}^{n_z} \|[W_j^1]_i\|_2, \tag{9}$$

where $\alpha$ is the penalty. In addition to the sparsity-promoting loss, we leverage a magnitude-based pruning method to capture the structure more explicitly. As the training proceeds, we prune a column whose $\ell_2$-norm becomes smaller than a certain threshold, $\rho$:

$$[W_j^1]_i = \boldsymbol{0} \quad \text{if} \quad \|[W_j^1]_i\|_2 \leq \rho, \tag{10}$$

where $\boldsymbol{0}$ is a zero-valued vector. By applying the sparsity-promoting loss and the pruning method, we expect that the entries of $\bar{z}$, that are non-causal to specific time-series, are pruned and, thus, the lags where dependencies exist can be identified automatically.

### 3.3 PUTTING ALL TOGETHER

In the implementation, we employ the standard mini-batching and a variant of stochastic gradient descent (SGD) to train the neural network architectures described in the previous subsections. With the sparsity promoting penalty and the pruning, the entire training process shares the commonality with the training algorithm proposed in Lee et al. (2021b), shown in Algorithm 1. We implement our method in PYTHON using a deep learning framework, PYTORCH (Paszke, et al, 2019). For the NODEs/NDDEs capability, we use the TORCHDIFFEQ library (Chen et al., 2018). We will use the notation $\boldsymbol{x}$ to distinguish the time-stepped-series as opposed to the continuous series $\boldsymbol{z}$.

## 4 EXPERIMENTS ON CHAOTIC ODES

We first showcase the effectiveness of the proposed method with two canonical chaotic ODE benchmark problems: the Lorenz-96 system and the Mackey–Glass equation. The Lorenz-96 system has been an important testbed for climate modeling. We use this system to showcase the structure learning in the context of NODEs (with the assumption that there is no delayed effect). The Mackey–Glass (MG) equation is another chaotic system, describing the healthy and pathological behaviour in certain biological contexts (*e.g.*, blood cells). We use MG to demonstrate the structure learning with *delayed* variables in the context of NDDEs. In the both experiments, we measure performance in two different angles, accuracy in trajectory reconstruction and structure discovery.

## 4.1 Benchmark 1: the Lorenz-96 system

The Lorenz-96 system (Lorenz, 1996) is given by the equation:

$$\frac{dx_i}{dt} = (x_{i+1} - x_{i-2})x_{i-1} - x_i + F, \tag{11}$$

where $i = 1, \ldots, N$ with $x_{-1} = x_{N-1}$, $x_0 = x_N$, and $x_1 = x_{N+1}$, $F$ denotes the forcing term.

**Setup.** For modeling, we use the component-wise NODE where each MLP, $f_i$, has 4 layers with 100 neurons, where $z_i$ models $x_i$ with $n_z = N$ in Eqn. 5. For the nonlinearity, we use the Tanh activation. We put the detailed experimental setup in Appendix.

**Results.** Fig. 1 reveals the learned dependence structure by showing the magnitude of the column norms of the weight matrices, $\{W_1^j\}$, in the input layer; the magnitude is normalized to have 1 as the maximum value (black) and 0 as the minimum value (white). The horizontal entries (*i.e.*, $\{x_i\}$) with darker colors (close to the black color) can be considered as the ones with the significant contributions to the vertical entries (*i.e.*, $\{\dot{x}_i\}$). Fig. 1 shows that the models have learned that $\dot{x}_i$ is dependent on $x_{i-2}, x_{i-1}, x_i, x_{i+1}$ and not dependent on other entries. Further experiments with varying $N$ can be found in Appendix. As finding structures in NODEs is not the main contribution in this work and the same benchmark problem has been studied in Bellot et al. (2022), we refer readers to Bellot et al. (2022) for more experimental results.

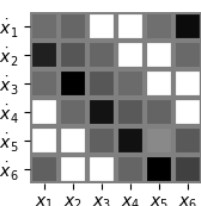

Figure 1: [Lorenz-96] Identified dependence structure for the systems with $N = 6$.

## 4.2 Benchmark 2: the Mackey–Glass system

The Mackey–Glass (MG) system (Mackey & Glass, 1977) is given by the equation:

$$\frac{dx}{dt} = -bx(t) + a\frac{x(t-\tau)}{1 + x(t-\tau)^c}, \tag{12}$$

where $a, b$ and $c$ denote the ODE parameters and $\tau$ denotes the lag. The values of $x(t)$ for $t \leq 0$ is defined by the initial function $\phi(t) = .5$.

**Setup.** For parameterizing NDDEs, we model the right-hand side of the NDDE as an MLP that takes $m = 10$ candidate delayed variables along with the current variable as an input, such that

$$\bar{\boldsymbol{x}}(t) = [x(t), x(t-\tau_1), \ldots, x(t-\tau_{10})] \in R^{11}, \tag{13}$$

where $\tau_i = i$ (seconds), for $i = 1, \ldots, 10$. We use an MLP consisting of 4 layers with 25 neurons with the Tanh activation function. We refer readers to Appendix for the detailed experimental setup.

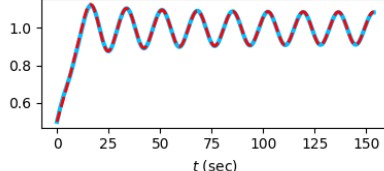

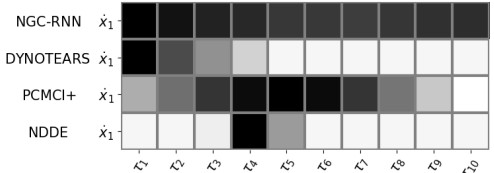

Figure 2: [Mackey–Glass] The ground-truth trajectory (solid blue) and the predicted trajectory computed from the learned model (dashed red).

Figure 3: [Mackey–Glass] Identified dependence structure for the systems with 10 delayed variables via four different methods: NGC-RNN, DYNOTEARS, PCMCI+, and NDDE.

**Results.** Fig. 2 depicts the ground-truth trajectory and the predicted trajectory computed from the learned NDDE model. We repeated the same training for five times with different initializations. Although the mean and the two standard deviation are plotted in Fig. 2, each prediction appear

almost identical. Fig. 3 depicts the learned dependence structure. The values are normalized to lie between [0, 1]. We can observe that the most significant contributes are from $x(t-\tau_4)$ and $x(t-\tau_5)$, meaning that the delayed effect appears within a 4~5-seconds window, correctly predicting true positive, but with one false positive error.

**Comparisons to baselines.** As baselines of comparisons, we consider the recent statistical and machine learning methods for identifying causalities in time-series data with lagged (i.e., delayed) variables: PCMCI+ (Runge, 2020), DYNOTEARS (Pamfil et al., 2020), and neural Granger causality (Tank et al., 2021) - RNN variants (NGC-RNN). We refer readers to Appendix for more descriptions on these methods. For all methods, we repeat three independent runs. Fig. 3 shows a heatmap of numerical values indicating the causal strength, which are normalized in a row-wise fashion (i.e., normalized per each method). The proposed NDDE-based method outperforms all other baselines. The second best performing method is the PCMCI+, which somewhat captures the strongest dependence from the fifth delayed variable $x(t - \tau_5)$ (true positive), but fails to find a correct structure, having many nonzero values in dependent elements (many false positives). The other two baselines find the strongest dependence on the first delayed variable. Moreover, the proposed method is the only method that results in accurate reconstruction of the ground-truth trajectory (Figure 2).

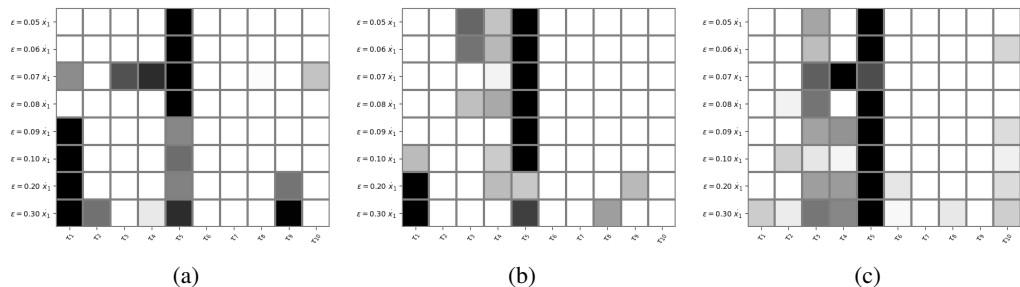

(a)  (b)  (c)

Figure 4: [Mackey–Glass] Identified dependence structures under increasing noise levels. (a) Noise added to trajectory. (b) Noise added to trajectory with anomalies. (c) Noise added to ODE (Ours).

**Resilience to Gaussian noise and anomalies.** We test the resilience of our approach to Gaussian noise and anomalies in three different contexts: (a) when the ground-truth trajectory is corrupted by additive Gaussian noise; (b) when the oracle trajectory contains Gaussian noise and 10 non-trivial anomalies (Thill et al., 2020), and (c) when we add noise to the velocity function of the ODE describing the Mackey–Glass system (Eqn. 12). (c) is the same approach considered in NGM (Bellot et al., 2022) to inject noise in the measurement. We describe the detailed setup in Appendix.

Although increasing noise level tend to negatively affect the performance of the proposed method (i.e., introducing more false positives), the method is successful in finding the true positive ($\tau = 5$) for all scenarios and all noise levels. In Fig. 4(a), our method learns the strongest dependence on the correct ground truth delay ($\tau = 5$) with up to around 8% additive noise. In Fig. 4(b), our method does the same up to 10% noise. In Fig. 4(c) the strongest dependence on the correct delay is learned at all noise levels. In the first two cases, adding low levels of noise appears to mitigate the 4~5 second discrepancy mentioned above. In all cases, the addition of noise appears to regularize the result. Particularly, the experimental setup and results in (c), i.e., adding noise to the ODE, shows the resilience of the dependence structures found by our method to all noise levels; the points of the trajectory are smooth while the trajectory itself is irregular due to perturbing evaluation of the system while running the ODE solver.

We also test the same baselines considered above, PCMCI+, DYNOTEARS, and NGC-RNN, on the noise/anomaly-perturbed measurements. All three baselines learn the dependence structures that are roughly the same as shown in Fig. 3; DYNOTEARS fails to capture the true positive dependence while PCMCI+ (NGC-RNN) produces many false positive errors (finding dependence on nearly all variables). We refer readers to Appendix G.2 for detailed results.

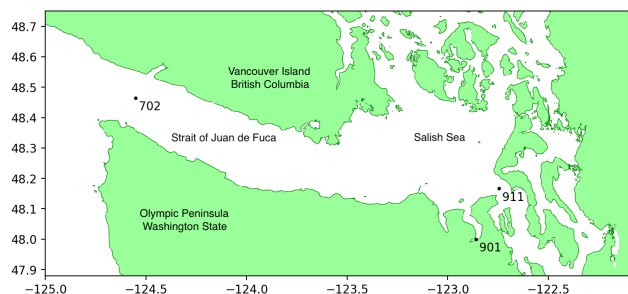 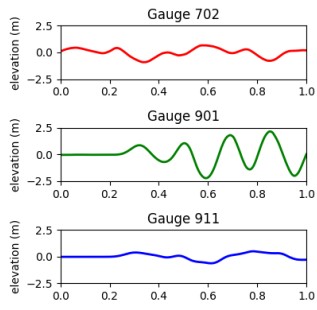

Figure 5: Tsunami simulation at the Strait of Juan de Fuca, depicting the location of Gauge 702, 901, and 911.

Figure 6: Surface elevation at Gauge 702, 901, and 911.

## 5 AN APPLICATION TO TSUNAMI FORECASTING

Now we further investigate the forecasting and structure discovery capabilities of the NDDEs equipped with the learning algorithm proposed in Sec. 3. We examine them with a dataset of tsunami realizations used for developing tsunami forecasting models (Liu et al., 2021). There exist only a handful of real earthquake events with large magnitudes, so synthetic tsunami data has been generated using numerical tsunami simulations that model the physics. In particular, the tsunami wave propagation modeled by nonlinear partial differential equations (LeVeque et al., 2011). The simulation is initiated by an incoming tsunami wave, which interacts nonlinear with the topography and reflecting waves. This particular setup makes the dataset ideal for interpreting the results.

### 5.1 TSUNAMI DATASET

The dataset was created from a set of 1300 synthetic Cascadia Subduction Zone (CSZ) earthquake events ranging in magnitude from Mw 7.8 to 9.3 (Melgar et al., 2016) and made available in Melgar (2016). These were generated from methods proposed in LeVeque et al. (2016) using the MudPy software (Melgar, 2020). The resulting seafloor deformation was then used as initial conditions for tsunami wave propagation implemented in Clawpack Development Team (2020).

The tsunami data for one earthquake event contains tri-variate time-series data with the variables $(\boldsymbol{x}^{(702)}, \boldsymbol{x}^{(901)}, \boldsymbol{x}^{(911)})$. The time series has duration of 5-hours and is interpolated at 256 uniformly spaced points on the time-grid. This time-series data corresponds to gauge readings for the synthetic tsunami entering the Strait of Juan de Fuca (SJdF). Each variable corresponds to different geological locations denoted by the number designations 702, 901, and 911, as shown in Fig. 5. Gauge 702 is located at the entrance of SJdF, and the other two gauges are located further inside from the entrance: Gauge 901 is located in Discovery Bay, and Gauge 911 is located in the middle of Admiralty Inlet. Fig. 6 shows an example of the surface elevation time-series measured at the three gauges.

Liu et al. (2021) considered tsunamis originating from hypothetical megathurst earthquakes in the CSZ that reach the Puget Sound. Since SJdF is the only path for the tsunamis to reach high-population areas in the Sound (Fig. 5), the authors hypothesized if observing the tsunami near the entrance of the strait (Gauge 702) could be used to forecast its amplitude at Gauge 901 and 911.

### 5.2 DESIGNING MODEL ARCHITECTURE

The base of our model is an NDDE, parameterized by using an MLP, that takes the three variables as the original input and the other three variables are their delayed versions, as follows:

$$\bar{\boldsymbol{x}}(t) = [\boldsymbol{x}^{(702)}(t), \ldots, \boldsymbol{x}^{(702)}(t-\tau_m), \boldsymbol{x}^{(901)}(t), \ldots, \boldsymbol{x}^{(901)}(t-\tau_m), \boldsymbol{x}^{(911)}(t), \ldots, \boldsymbol{x}^{(911)}(t-\tau_m)], \quad (14)$$

and outputs the time-derivatives of the three variables $\dot{\boldsymbol{x}} = \left[ \frac{d\boldsymbol{x}^{(702)}}{dt}, \frac{d\boldsymbol{x}^{(901)}}{dt}, \frac{d\boldsymbol{x}^{(911)}}{dt} \right]$.

We consider an MLP with 4 layers with 100 neurons and Tanh nonlinearity for each output element. As in the empirical study (Gusak et al., 2020) of applying different normalization techniques

to NODEs, we examined several combinations of normalization techniques, including layer normalization (LN) (Ba et al., 2016), weight normalization (WN) (Salimans & Kingma, 2016), and spectral normalization (Miyato et al., 2018). We found the best working configuration for this particular case study is [Linear→WN→LN→Tanh] × 4. We test 9 combinations of the weight penalty $\alpha \in \{0.1, 0.01, 0.001\}$ and the pruning threshold $\rho \in \{0.1, 0.01, 0.001\}$. We found that $\alpha = \rho = 0.01$ yields the best result. Promoting strong sparsity (larger $\alpha$ and $\rho$) or weak sparsity (smaller $\alpha$ and $\rho$) resulted in degradation in forecasting accuracy. We use this setting for our experiments.

## 5.3 TSUNAMI FORECASTING PERFORMANCE

To evaluate, we train NDDEs with varying number of delayed variables and varying time lags on the tsunami dataset and measure those models' forecasting accuracy as the relative $\ell_2$ error: $\|\boldsymbol{x}^{(\text{pred})} - \boldsymbol{x}\|_2 / \|\boldsymbol{x}\|_2$, where $\boldsymbol{x}^{(\text{pred})}$ and $\boldsymbol{x}$ denote the predicted and the ground-truth trajectories, respectively. We repeat the experiments 5 times with different initialization of model parameters.

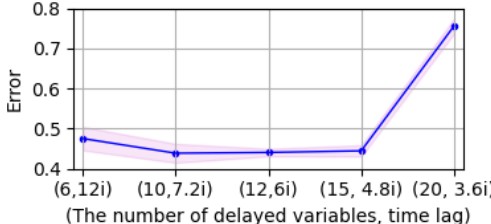

**Results.** We vary the value of the number of delayed variables and the time lag $(m, \tau_i) \in \{(6, 12i), (10, 7.2i), (12, 6i), (15, 4.8i), (20, 3.6i)\}$ while fixing the maximum time lag as 72 minutes. Introducing more delayed variables to the model only changes the input layer of the MLP and the specification of the internal layers is fixed. Fig. 7 shows the relative $\ell_2$ errors for varying $(m, \tau_i)$ and that increasing the number of delayed variables tends to decrease the error up to a certain number, i.e, 12 or 15. However, $m$ larger than 15 resulted in degradation of prediction accuracy.

Figure 7: [Tsunami] Averaged relative $\ell_2$ errors (blue line) for varying $m$ (the number of delayed variables) and time lag $\tau_i$. The magenta area is covered by the two standard deviations.

**Comparison to existing baselines.** We also consider classical deep-learning approaches for time-series modeling and the most recent approaches in tsunami forecasting on the same dataset (Liu et al., 2021): (1) classical DL approaches: recurrent neural networks (RNNs), long short term memory (LSTM) (Hochreiter & Schmidhuber, 1997), and gated recurrent unit (GRU) (Cho et al., 2014), (2) the state-of-the-art Tsunami forecasting approaches (Liu et al., 2021): denoising autoencoder (DAE) and variational autoencoder (VAE), (3) a variant of the

Table 1: [Tsunami] Performance comparisons

| MODEL | REL. $\ell_2$ ERROR | MODEL SIZE |
|---|---|---|
| RNN | $0.968 \pm 0.0213$ | 219K |
| LSTM | $0.984 \pm 0.0042$ | 875K |
| GRU | $0.909 \pm 0.0101$ | 656K |
| DAE | $0.474 \pm 0.0265$ | 2,604K |
| VAE | $0.461 \pm 0.0111$ | 3,297K |
| NODE | $1.061 \pm 0.0085$ | 95K |
| ANODE | $1.104 \pm 0.0206$ | 93K |
| NDDE (ours) | $0.439 \pm 0.0118$ | 107K |
| LI-NDDE (ours) | $0.404 \pm 0.0087$ | 163K |

proposed method: latent-input augmented NDDEs (LI-NDDEs). The LI-NDDEs (1) extract a latent code $\boldsymbol{c}^{(k)}$ from the $k$th data instance using a small-sized neural network (see model size in Table 1) and (2) augment it to the state $\boldsymbol{z}$ (i.e., $\boldsymbol{f}([\boldsymbol{z}, \boldsymbol{c}^{(k)}], t; \boldsymbol{\Theta})$. We include more details in Appendix.

Table 1 shows the relative $\ell_2$ errors obtained by using all baselines with the best performing hyperparameters (see Appendix for hyperparameter settings). The results essentially show that the our models outperforms the baseline in terms of prediction accuracy while keeping the model size small.

## 5.4 DEPENDENCE STRUCTURES

In Fig. 8, we report the learned dependence structure. The magnitude of the column norms of the weight matrices $\{W_j^1\}$ in the input layer are again considered as the amount of contributions made by the corresponding input variables. We repeat the same experiments 15 times with different initialization, obtain the column norms from each run, and then apply the kernel density estimation based on Scott's factor (Scott, 2015) to the collected column norms.

We first focus on the plots on the diagonal of the $3 \times 3$ plot matrix. Gauge 702 has only small contributions from the delayed variables. The gauge observes the incoming wave and there is little reflections coming from the interior of the Sound, so it is expected that there are only small contri-

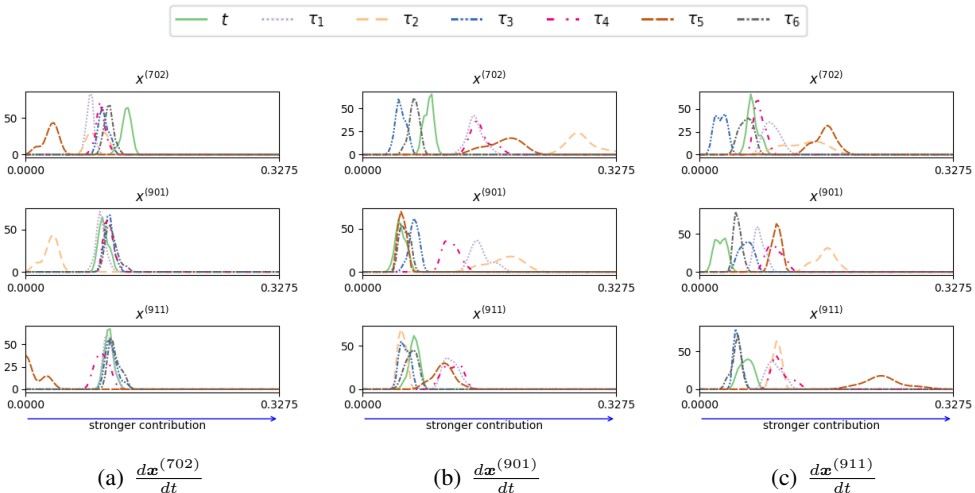

Figure 8: [Tsunami] Dependence structures of NDDE models with $m = 6$: the contributions from $\boldsymbol{x}^{(702)}$ (top), $\boldsymbol{x}^{(901)}$ (middle), and $\boldsymbol{x}^{(911)}$ (bottom) to $\dot{\boldsymbol{x}}^{(702)}$ (left), $\dot{\boldsymbol{x}}^{(901)}$ (middle), and $\dot{\boldsymbol{x}}^{(911)}$ (right).

butions to the result. Gauges 901 and 911 have higher contributions since they do possess dependent behavior coming from the reflections caused by the narrowing strait. This is especially pronounced for 901 which sits in Discovery bay and experiences significant sloshing due to the local topography (see Figs. 6 and 17). Next, we observe that the upper diagonal plots reveal larger contributions where as the lower-diagonal plots do not. This indicates that the delayed time-series of Gauge 702 causes those of Gauge 901 and 911 in a significant way, and that delayed time-series at Gauge 901 causes Gauge 911. The significant contributions agree well with the spatio-temporal progression of the physical wave itself.

The most significant delayed variables from Gauge 702 in predicting Gauge 901 are the delays $\tau_2$ and $\tau_5$, which correspond roughly to 24 minutes and 60 minutes of delay. Thus 30-60 minutes of the time-series from 702 is required to predict the time-series at Gauge 911, which agrees with the prediction results from Liu et al. (2021): while 30 minutes of time-series from Gauge 702 was sufficient to predict the wave height at Gauge 901, using 60 minutes improved the result significantly.

We also use the DYNOTEARS (Pamfil et al., 2020) and PCMCI+ (Runge, 2020) APIs to find dependence structures. The both methods produce the similar results that the strongest dependence on Gauges 901 and 911 is from Gauge 702 with $\tau_6$, which is $\sim$12 min after the wave reaches the Gauges 901 and 911, while not showing any significant dependence from other $\tau_i$, suggesting that the result do not match with the spatio-temporal progression of the physical wave.

## 6  CONCLUSION

We propose the computational framework for time-series forecasting and structure discovery from continuous-time dynamics models with delayed variables. To this end, we adapt score-based structure learning algorithms to delay dynamics settings and develop a training algorithm that promotes sparse dependence structure in the parameter space. We evaluate our method with numerical experiments on two chaotic-ODE benchmark problems: the Lorenz-96 and Mackey–Glass equations. We also showcase our method's capabilities in (1) accurate dynamics modeling and (2) structure discovery, and (3) resiliency to additive noise and anomalies. We finally present an application of our method to tsunami forecasting. Our method produces highly accurate tsunami forecasting (*i.e.*, the predictions match the highest peaks of sea-surface elevation) and also identifies the dependence structure of the time series obtained from the three gauges, which are physically-consistent and agree with the domain expert's interpretation.

ETHICS STATEMENT

There is no known ethics issue according to the ICLR Code of Ethics.

REPRODUCIBILITY

To ensure the reproducibility, we provide the details of the experimental setups including model architectures and hyperparameters in Appendix E. Along with this information, we also provide our code as a supplemental material.

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

## A   GRAPHICAL ILLUSTRATION OF DEPENDENCE STRUCTURE LEARNING

Figure 9 illustrates examples of learned dependence structure: illustrating example results of NGM in NODEs (Figure 9(a)) and NGM in NDDEs (Figure 9(b)).

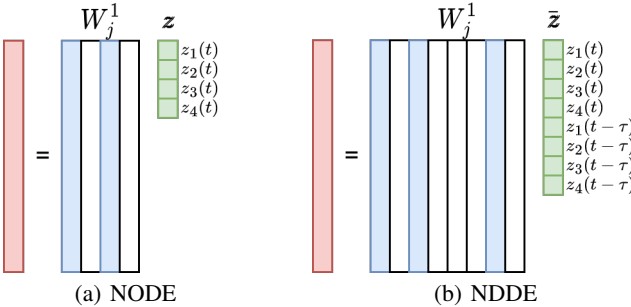

(a) NODE                    (b) NDDE

Figure 9: An illustrative operation of the input layer (the white bars indicate zero-norm columns in weights).

## B   ANODES AND ANDDES

We provide the background knowledge of augmented neural ordinary differential equations (AN-ODE) or neural delay differential equations (ANDDEs). ANODEs lift NODEs by adding arbitrary extra dimensions in the state variables leading to a system of ODEs, formally expressed as follows:

$$\begin{bmatrix} \dot{z} \\ \dot{s} \end{bmatrix} = f\left(\begin{bmatrix} z \\ s \end{bmatrix}; \Theta\right). \tag{15}$$

Even if we have not numerically tested ANDDEs in this study, it is a straightforward extension from NDDEs to ANDDEs. We can augment NDDEs to take extra input dimensions as follows:

$$\begin{bmatrix} \dot{z} \\ \dot{s} \end{bmatrix} = f\left(\begin{bmatrix} \bar{z} \\ s \end{bmatrix}; \Theta\right) \tag{16}$$

where the initial function of the augmented variables defined as $s(t) = [s_1(t), \ldots, s_{n_s}(t)]^\mathsf{T} = \mathbf{0}$ if $t \leq 0$. This is a straightforward extension of ANODEs in NDDEs.

## C   INPUT-AUGMENTED NODES AND LATENT-INPUT AUGMENTED NODES

An input-augmented NODEs (Massaroli et al., 2020) is an extension of NODEs, which augments the input to the hidden latent states, e.g., $f([z, X^{(k)}], t; \Theta)$, where $X^{(k)}$ is the $k$th data instance. As this work aims to perform forecasting, augmenting the input (i.e., the entire time-series) is not feasible. Thus, we extract a hidden feature by using an autoencoder, whose input is again the first $n_{\text{in}}$ steps of the Gauge 702, and augmented the extracted latent code $c$ to the dynamics, i.e., $f([z, c^{(k)}], t; \Theta)$, which we denote by latent-input-augmented (LI)-NDDEs. The autoencoder used here has the same input/output setting as the previously described baseline autoencoders, but is in a small scale (see Table 1 for the model size).

## D   DESCRIPTIONS ON BASELINES

- DYNOTEARS: is a scored-based algorithm for learning Dynamic Bayesian Networks (DBNs) that simultaneously discover the contemporaneous and time-lagged relationships among variables in a time-series. The model follows the standard SVAR model and uses standard opti-

mization routines to minimize a penalized loss subject to an acyclicity constraint (Pamfil et al., 2020). We use CAUSALNEX [1] API.

- PCMCI+: is a three-phase conditional independence (CI) based algorithm that aims to discover the contemporaneous and lagged causal from observational time series. By utilizing the Peter–Clark (PC) algorithm and the Momentary Conditional Independence (MCI) test, PCMCI+ optimizes the choice of conditioning sets in CI tests to achieve high detection capability while maintaining well-calibrated tests (Runge, 2020). We use TIGRAMITE [2] API.

- Neural Granger causality (NGC): proposed to adapt neural network architectures in a way that the Granger causal interactions (Granger, 1969) (or neural Granger causality) are estimated while performing data-driven dynamics modeling of nonlinear systems. Component-wise MLPs or RNNs are leveraged to capture the effects of input on individual output series.

## E  EXPERIMENTAL SETUP IN DETAIL

**Lorenz-96 system.** In Sec. 4.1, we set the forcing term as $F = 10$, which causes a chaotic behavior. The initial condition is set as $\boldsymbol{x}(0) = [1, 8, \ldots, 8]$, perturbed by adding a noise which is sampled from uniform distribution $U(-1, 1)$. The ground-truth trajectories are generated by solving the initial value problem with the time-step $\Delta t = 0.01$ for the total simulation time $T = 30.72$. For the time integrator, we use the Dormand–Prince method (dopri5) (Dormand & Prince, 1980) with the relative tolerance of $10^{-7}$ and the absolute tolerance of $10^{-9}$. We set the training hyperparameters as follows:

- Learning rate: 0.01, max epoch: 2000, batch size: 40, and batched subsequence length: 100.

**Mackey–Glass system**. In Sec. 4.2, we set $a = 0.2$, $b = 0.1$, $c = 10$, and $\tau = 5$ (seconds). We solve the initial value problem, given the initial function $\phi(t) = .5$, using DDEINT[3] with the time-step $\Delta t = 0.01$ for the total simulation time $T = 153.6$ (seconds). We set the training hyperparameters as follows: for NDDEs,

- learning rate: 0.01, max epoch: 2000, batch size: 40, batched subsequence length: 100, and ODE integrator: Runge–Kutta (Runge, 1895) of order 4.

For RNN variants,

- learning rate: 0.001, max epoch: 1000, batch size: 40, batched subsequence length: 153, and penalty weight 0.001 for RNN, 0.01 for LSTM, and 0.1 for GRU.

**Tsunami forecasting** We set the training hyperparameters as follows: for NDDEs,

- learning rate: 0.001, max epoch: 1000, batch size: 40, batched subsequence length: 10, and and ODE integrator: Runge–Kutta of order 4.

For RNN variants,

- learning rate: 0.001, max epoch: 1000, batch size: 40, batched subsequence length: 19, and penalty weight 0.1 for RNN, 0.1 for LSTM, and 0.01 for GRU.

For autoencoders, we take the same hyperparameters from Liu et al. (2021): the model specifications are

- [DAE] an encoder: 8 convolutional layers with kernel size 3 and channels [64, 64, 128, 128, 256, 256, 512, 512] and a decoder: 8 transposed-convolutional layers with kernel size 3 and channels in the reverse order of the encoder,

---

[1]`https://github.com/quantumblacklabs/causalnex/blob/develop/causalnex/structure/dynotears.py`
[2]`https://github.com/jakobrunge/tigramite/blob/master/tigramite/pcmci.py`
[3]`https://pypi.org/project/ddeint`

- [VAE] an encoder: the same specifications for convolutional and transposed-convolutional layers, an additional fully-connected (FC) layers to/from the latent code of dimension 450.

and hyperparameters are

- [DAE] loss: L1 loss, learning rate: 0.0005, max epoch: 750, and batch size 20, and
- [VAE] loss: ELBO, learning rate: 0.0005, max epoch: 1250, and batch size 150.

For the latent-input-augmented NDDEs, the small-scale autoencoder is specified as:

- an encoder: 4 convolution layers with kernel size 3 and channels [8,16,32,64] and one FC layer to the latent code of dimension 5, and a decoder: one FC layer and 4 transposed-convolutional layers with kernel size 3 and channels in the reverse order of the encoder,

and hyperparameters are:

- loss: mean-squared error, learning rate: 0.001, max epoch: 1000, and batch size 50.

We also note that in the recent Tsunami forecasting approaches, the autoencoders have been designed in a non-typical way; the input to the autoencoders is the first $n_{in}$ steps of the Gauge 702 time-series and the output is the next $256 - n_{in}$ steps of all time-series (i.e., 702, 901, and 911). In our experimental settings, we set $n_{in} = 60$ (i.e., 72 min) to be consistent with the settings of NDDEs.

## F  LORENZ-96 EXPERIMENTAL RESULTS

Figure 10 depicts the ground-truth trajectories and the predicted trajectories that are computed via solving the IVP given the learned model. We observe that the predictions match well with the ground-truth trajectories.

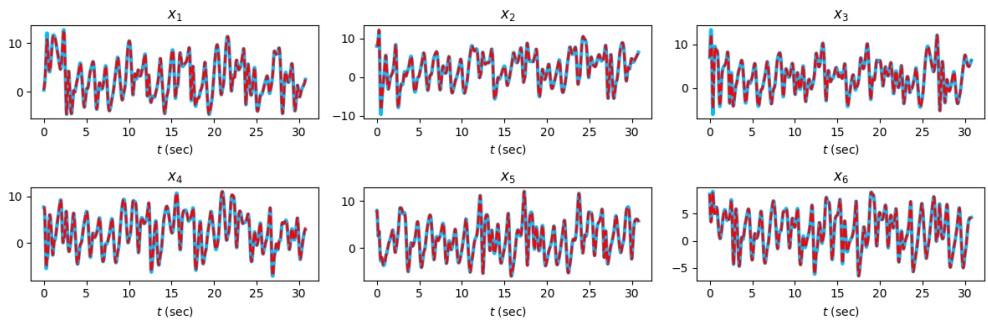

Figure 10: [Lorenz-96] The ground-truth trajectories (solid blue) and the predicted trajectories computed from the learned model (dashed red).

Figure 11 depicts the identified dependence structure for varying $N = \{6, 8, 10\}$; the true positives are all correctly captured by the algorithm.

## G  MACKEY–GLASS: FULL RESULTS FROM TESTING THE RESILIENCE TO NOISE

We train NDDEs on noisy Mackey–Glass system using the same ODE parameters and model hyper-parameters as the baseline above. For all our experiments, we use a set of increasing scaling factors $\epsilon = \{.05, .06, .07, .08, .09, .1, .2, .3\}$, with fine-grained steps between .05 and .1. Throughout, we use a pre-computed standard deviation $\sigma_{base}$ taken from the original trajectory. We add noise using three separate methods:

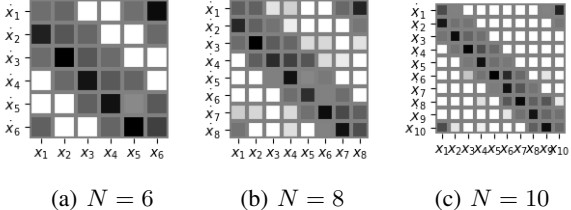

(a) $N = 6$        (b) $N = 8$        (c) $N = 10$

Figure 11: [Lorenz-96] Identified dependence structure for the systems with $N = 6, 8, 10$.

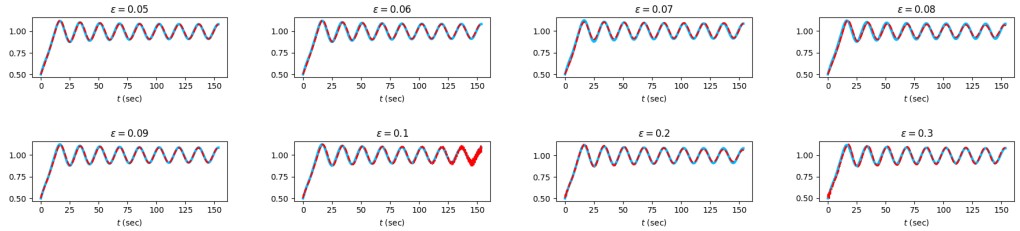

Figure 12: [Mackey–Glass] The original, ground-truth ($\epsilon = 0$) trajectory (solid blue) and predicted trajectories (dashed red) when noise-level $\epsilon$ is added to the original trajectory.

(a) Following the prior work (Lee et al., 2021b), we sample noise from a Gaussian distribution $N(0, \sigma_{base}^2)$, multiply it by a varying scaling factor $\epsilon$, and add it to the original trajectory.

(b) We use MGAB (Thill et al., 2020) to produce a single benchmark with 10 non-trivial anomalies. MGAB removes random segments from time series data such that the remaining endpoints have closely matching derivatives. This results in anomalies which appear smooth and are difficult to detect visually. We add Gaussian noise to this benchmark according to $\epsilon$ as above.

(c) We sample from a Gaussian distribution with standard deviation $\sigma_{base}$, scale by $\epsilon$, and add the result to the MG system (12) at each evaluation by the ODE solver. The ODE solver produces trajectories for the following.

$$\frac{dx}{dt} = -bx(t) + a\frac{x(t - \tau)}{1 + x(t - \tau)^c} + \epsilon N(0, \sigma_{base}^2), \tag{17}$$

where $\epsilon N(0, \sigma_{base}^2)$ can be described as a stochastic forcing term.

### G.1 NDDE RESULTS

Fig. 12 shows the predicted trajectories from the learned model for every noise level using this method. The predictions largely follow the ground-truth trajectories while appearing to slightly over-fit at higher noise levels in the places where the dashed red line becomes dense.

Fig. 13 shows the predicted trajectories against the anomalous data. Note that we include results for $\epsilon = 0$ as a baseline. The predictions again largely follow the $\epsilon = 0$ trajectory with the exception of the $\epsilon = .2$ case which begins to diverge at around $t = 75$. This pattern of deviation was observed during training as well, indicating that additional training could lead to more uniform trajectories.

Fig. 14 shows the predicted trajectories against the "irregular" trajectories generated using this method. The predicted trajectories appear to follow the irregular ones closely. In particular, the $\epsilon = .3$ case suggests the prediction more closely resembles the original baseline rather than the irregular trajectory.

### G.2 BASELINE RESULTS

Figs. 15– 16 show the causal strength of three baseline methods, NGC-RNN, DYNOTEARS, and PCMCI+, under different methods for injecting noise: the methods (a) and (c) described above.

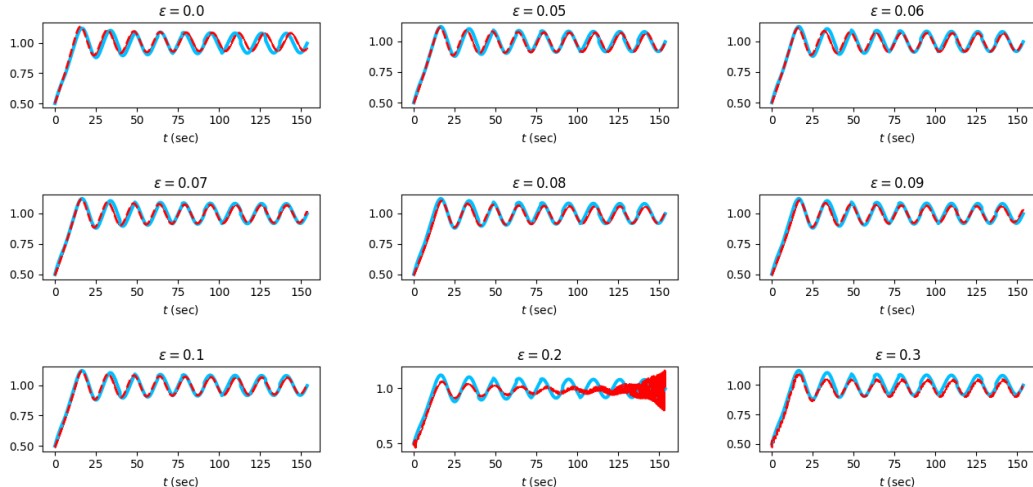

Figure 13: [Mackey–Glass] The anomaly benchmark trajectory (solid blue) and predicted trajectories (dashed red) when noise-level $\epsilon$ is added to the anomaly benchmark trajectory.

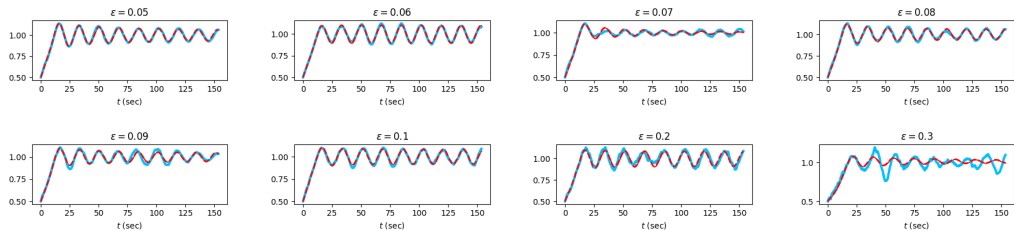

Figure 14: [Mackey–Glass] The "irregular" trajectories (solid blue) and predicted trajectories (dashed red) when noise-level $\epsilon$ is added to the ODE.

Figs. 15(b)–16(b) depict that DYNOTEARS fails to capture the correct dependence structure (true positive); instead it finds the strongest dependence on the first delayed variable. Figs 15(a)–16(a) and Figs. 15(c)– 16(c) show the results of NGC-RNN and PCMCI+, respectively, depicting that they perform poorly in finding a correct structure (i.e,. resulted in many false positives); although PCMCI+ finds the true positive correctly, it fails to detect non-dependence elements (leading false positive errors) and NGC-RNN in several cases even fail to detect true positive.

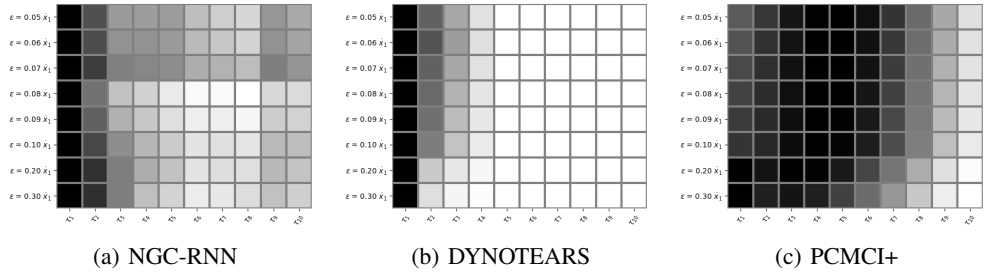

(a) NGC-RNN         (b) DYNOTEARS         (c) PCMCI+

Figure 15: [Mackey–Glass] Identified dependence structure under increasing noise levels added to trajectory via three different baseline methods: NGC-RNN, DYNOTEARS, and PCMCI+.

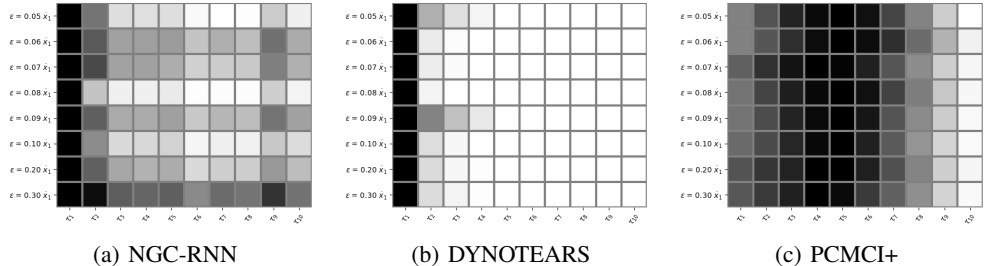

(a) NGC-RNN    (b) DYNOTEARS    (c) PCMCI+

Figure 16: [Mackey–Glass] Identified dependence structure under increasing noise levels added to ODE via three different baseline methods: NGC-RNN, DYNOTEARS, and PCMCI+.

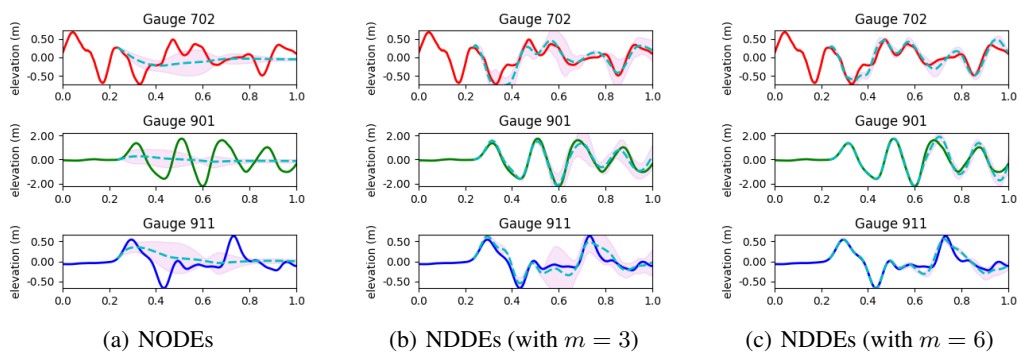

(a) NODEs    (b) NDDEs (with $m = 3$)    (c) NDDEs (with $m = 6$)

Figure 17: [Tsunami] An example of prediction results of using the trained NODEs and NDDEs.

## H   TSUNAMI EXPERIMENTAL RESULTS

### H.1   DATASET

Among the 1300 time-series instances of tsunami, the events that with negligible tsunami in the region of our interest were discarded; event with tsunami amplitudes less than 0.1m at Gauge 702 or 0.5m at Gauge 901 were removed from the dataset. This leads to the decrease in the number of time-series instances in the dataset from 1300 to 959. We then split the remaining data into 80/5/15 for the train/validation/test set.

### H.2   NODE RESULTS

Fig. 17(a) depicts an example instance of the elevation time series in the test set and the prediction made by solving an IVP given the trained NODE model and the initial condition obtained from the test set. We report the mean of the predictions (cyan dashed line) and the two standard deviations (magenta area). The trained NODEs can only produce simple trajectories, which do not forecast the oscillatory behavior in the data and fail to match the peak values and the locations in the time series. We also test ANODEs, but ANODEs do not provide significantly different results than NODEs (see Appendix).

### H.3   NDDE RESULTS

**With the fixed $\tau$ and varying $m$.**   We test the NDDEs with varying number of delayed variables (*i.e.*, $m \in \{1, 2, 3, 4, 5, 6\}$) where $\tau_i = 12i$ (min), i.e., the time lag is defined as 12 minutes. As depicted in Fig. 18, the increase in the number of delayed variables tends to result in better performance in terms of prediction accuracy measured in mean-squared errors.

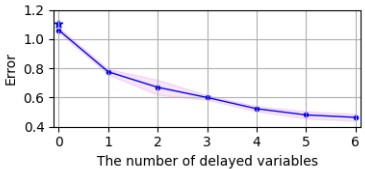

Figure 18: [Tsunami] Averaged relative $\ell_2$ errors for varying $m$ (the number of delayed variables) and the fixed time lag $\tau_i = 12i$. Star marker indicates the error of ANODE results.

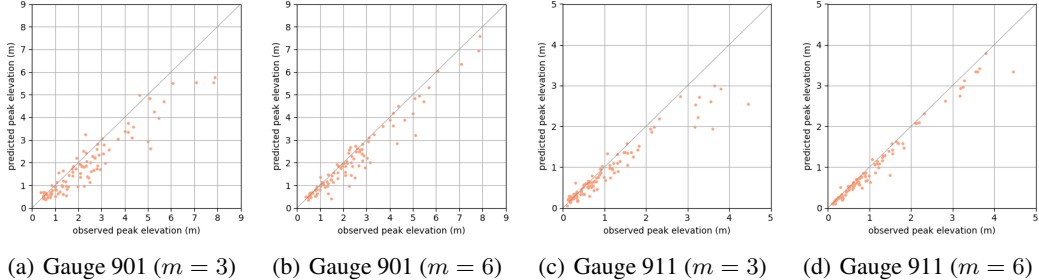

(a) Gauge 901 ($m = 3$) (b) Gauge 901 ($m = 6$) (c) Gauge 911 ($m = 3$) (d) Gauge 911 ($m = 6$)

Figure 19: [Tsunami] Observed peak locations (horizontal axis) and predicted peak locations (vertical axis) for elevation time-series measured at Gauge 901 and 911.

The figure also shows the elevation time series for the same test data considered in the experiments with NODEs. We plot the mean of the predictions as cyan dashed lines and the two standard deviations (as magenta areas). We use the same experimental procedure for forecasting (*i.e.*, solving IVP with the learned NDDEs). As opposed to the predictions made by NODEs, the predictions made by NDDEs match well with the ground-truth trajectories and capture the peak values and locations more accurately. Fig. 19 show the predicted peak values more aligned with the ground-truth trajectories at Gauge 901 and 911. Two sets of the predictions made separately by NDDEs with $m = 3$ and $m = 6$ are depicted. The NDDEs with larger $m$ provide better predictions for peak values.

