# OpenReview forum: "Promoting Sparsity in Continuous-time Neural Networks to Learn Dependence Structures"
_ICLR.cc/2024/Conference — ICLR 2024 Conference Withdrawn Submission_

### Official Review · Reviewer_EHcX · 2023-10-31

**Soundness:** 2 fair
**Presentation:** 3 good
**Contribution:** 1 poor
**Rating:** 3
**Confidence:** 3

**Summary:**

The paper introduces a penalization and a pruning strategy to promote the sparsity of neural network parameters, with the aim of enhancing the model forecasting accuracy and interpretability for dependence identification.

**Strengths:**

1. the paper is well-written, with a comprehensive background and a clear presentation of motivations, model architectures, and results.
2. Extensive simulation and real data experiments are included in the paper, helping the readers better understand the model benefits.

**Weaknesses:**

1. The contribution of the paper seems to be incremental. If I understand right, the paper only explores neural networks with one layer. If that is the case, I expect some theoretical results of the proposed method, which I haven't found yet.
2. On the other hand, the previous work [1] has presented a detailed analysis of such a neural model in the problem of graphical modeling of ODE, which makes me question the contribution of this paper.
3. The technique of penalizing and pruning, without any theoretical proof, seems to be ad-hoc. More empirical or theoretical explorations on different penalizations/pruning would be crucial.

---
[1] Alexis Bellot, Kim Branson, and Mihaela van der Schaar. Neural graphical modelling in continuous time: consistency guarantees and algorithms. In International Conference on Learning Representations, 2022.

**Questions:**

1. Has the problem of the graphical modeling for NDDE been explored or proposed in previous work? I am not an expert in this area, but it would be helpful if the authors could provide some literature about this problem.
2. What if I use a parametric model, which could be more interpretable for dependency identification? Will the forecasting performance drop significantly? That will highlight the necessity of considering neural networks in this graphical modeling problem.

---

### Official Review · Reviewer_eu4v · 2023-10-31

**Soundness:** 2 fair
**Presentation:** 2 fair
**Contribution:** 2 fair
**Rating:** 3
**Confidence:** 4

**Summary:**

The paper leverages Neural Delayed Differential Equations (NDDEs) as a technique for identifying dependency structures in time series datasets. Distinct from previous methodologies, NDDEs possess the capability to learn and represent delayed relationships within the data. The approach is integrated within the established framework of Neural Graph Machines (NGM), leveraging its primary training architecture and regularization techniques. The efficacy of NDDEs is evaluated through experiments on two synthetic datasets and one application in tsunami forecasting. The results from these evaluations show the method’s superior performance in discovering underlying dependencies.

**Strengths:**

This paper tries to address the problem of structure learning in continuous time with delayed effects. The intuition and methodology is easy to follow. The author evaluates the proposed method not only on the synthetic data but also using the real-world tsunami forecasting dataset.

**Weaknesses:**

There are two primary concerns related to the proposed methodology. The first pertains to the novelty and contribution of the work. There is a significant overlap with prior work, specifically the Neural Graph Machines (NGM) framework. The similarities extend across various aspects, including the training loss, model architecture, and the implementation of sparsity regularizations. Notably, NGM encompasses additional features, such as adaptive lasso constraints, which are absent in the current proposal. The primary distinction between the two methodologies appears to be the substitution of Neural Ordinary Differential Equations (NODEs) with Neural Delayed Differential Equations (NDDEs), a modification that seems relatively straightforward. Furthermore, the approach to formulating the dependency graph mirrors that of NGM. Collectively, these observations lead to the judgement that the contribution of the current work is incremental and the integration of components is trivial.

The second concern centers on the theoretical underpinnings of the proposed method. A notable strength of NGM lies in its theoretical consistency guarantees pertaining to structure learning. In contrast, this paper does not engage in a discussion of theoretical guarantees for the proposed NDDE-based approach. While it may be plausible that the consistency guarantees of NGM are extendable to NDDE, the absence of a detailed discussion leaves uncertainties regarding key aspects such as consistency, identifiability, and the reliability of the learned structures. Addressing these theoretical considerations is important to validate the approach and ensure its robustness in capturing the true underlying data structures.

**Questions:**

1. What is $||f'_jz_i||$ ? Is this the derivative defined in the local dependency structure?
2. In the bottom of page 3 above Eq.8, shouldn't it be $\frac{dz_j}{dt}=f_j$?
3. In equation 9, what is $L$? It is not defined in previous context.
4. You should define $\tilde{z}$ in the main text rather than in the algorithm.
5. For equation 11, better to use $z$ instead of $x$
6. Considering the Rhino[1] as a baseline? It also supports lagged effect and instantaneous effect.
7. Is your method only applicable to single multi-dimensional time series? Since I think it may fail for multiple time series setting. For example, the ground truth data generating model is a stochastic differential equation (SDE) $dz = zdt + dW$, then we can simulate multiple time series from this SDE. However, if you only use NODE to learn the mean process, then the optimal drift is 0, which cannot recover the ground-truth trajectory.


[1] Gong, Wenbo, et al. "Rhino: Deep causal temporal relationship learning with history-dependent noise." arXiv preprint arXiv:2210.14706 (2022).

---

### Official Review · Reviewer_tEwg · 2023-11-02

**Soundness:** 2 fair
**Presentation:** 2 fair
**Contribution:** 2 fair
**Rating:** 5
**Confidence:** 3

**Summary:**

This paper modifies the NGM model from Bellot 2022 to incorporate Neural delay differential equations from Zhu et al 2020, with an additional sparsity promoting prior for interpretable dependence structure learning from time series data.

**Strengths:**

The problem of interpretable structure learning from time series data is an interesting question relevent to many time series datasets. Neural graphical models can capture causal dependence structures and seems to be suitable for introducing time-delay via the analogy with NDDs to NODEs.

**Weaknesses:**

The modifications seem incremental, no properties of their method are explored, the methods section simply proposes the merging of the NDD with the NGM idea. the experimental results can be improved by
1. Comparing to GODE models like Ritini from Bhaskar et al. 2023.
2. Other forms of structure learning like transfer entropy or granger causality via a neural network.

The chaotic dynamics performance evaluation is a bit shaky. The marginal improvements on these datasets does not indicate actual learning of stochastic dynamics, but potentially an overfitting. I would suggest more patterned datasets.

Also while delay is modeled here hysteresis which is apparent in biological systems is not modeled.

**Questions:**

It is not clear if the delay in inputs is learned, a hyperparameter or fixed? In other words are you expecting the lag to be known because this is not usually the case.

---

### Official Review · Reviewer_3bqx · 2023-11-02

**Soundness:** 2 fair
**Presentation:** 2 fair
**Contribution:** 2 fair
**Rating:** 3
**Confidence:** 4

**Summary:**

This paper proposes a method for pruning input-layer weights in neural networks, which are trained to approximate time-dependent dynamics from time-series data. The core concept involves applying group lasso regularisation to the input layer parameters, with each group representing sets of weights connecting individual input-layer neurons to the first hidden layer. During training, as the weights are updated through batch updates in stochastic gradient descent, a group of parameters is set to zero if the l2 norm of the group falls below a specified threshold, which serves as a hyper-parameter for the method. According to the authors, a pruned-out group of weights indicates that the corresponding input neuron is independent of the underlying temporal dynamics. Thus, the method can be employed to identify interpretable temporal dependence structures in time-series data. Numerical experiments are conducted using both simulated and real data. The results demonstrate that for univariate simulated data, the proposed approach is capable of accurately identifying the true dependence structure. In comparison, other comparable approaches either fail to discover the true dependencies or identify numerous false positives. For real data, the authors evaluate predictive accuracy and draw interpretations regarding the consistency of the discovered dynamics with the understanding of the physical process.

**Strengths:**

The paper is well-written and easy to follow.

The proposed pruning step is simple yet effective in identifying instantaneous and time-lagged dependencies, as demonstrated in experiments on simulated data.

The approach outperforms various sophisticated and state-of-the-art methods, some of which exhibit substantial shortcomings in identifying the ground truth or missing it completely.

**Weaknesses:**

While applying group lasso to the discretised form of time-dependent differential equations may be novel, the concept of a group penalty is not. Therefore, the overall novelty and significance of the work is limited.

The impact of model architecture and input characteristics on the effectiveness of the approach is not clearly addressed. The experiments presented in the paper are of a small scale and do not thoroughly explore this aspect.

The method follows a greedy approach, potentially selecting one correlated variable based on initial conditions and pruning out others as irrelevant. This may lead to false negatives.

The experiments mainly focus on univariate time-series, thus lacking investigation of the method's effectiveness in the high-dimensional multivariate (and noisy) case where interdependencies between dimensions may exist. Extending the synthetic dataset experiments to large-scale multidimensional time-series could provide a valuable benchmark comparison for the proposed method and other approaches.

**Questions:**

Can you please clarify if the results reported in Table 1 are based on a held-out segment of data?